# Acetylome and Succinylome Profiling of *Edwardsiella tarda* Reveals Key Roles of Both Lysine Acylations in Bacterial Antibiotic Resistance

**DOI:** 10.3390/antibiotics11070841

**Published:** 2022-06-23

**Authors:** Yuying Fu, Lishan Zhang, Huanhuan Song, Junyan Liao, Li Lin, Wenjia Jiang, Xiaoyun Wu, Guibin Wang

**Affiliations:** 1School of Safety and Environment, Fujian Chuanzheng Communications College, Fuzhou 350007, China; 2021003@fjcpc.edu.cn (Y.F.); Junyanliao@126.com (J.L.); linli37@mail2.sysu.edu.cn (L.L.); jiangwenjia23@163.com (W.J.); juju0928@163.com (X.W.); 2Fujian Provincial Key Laboratory of Agroecological Processing and Safety Monitoring, School of Life Sciences, Fujian Agriculture and Forestry University, Fuzhou 350002, China; 2200525002@fafu.edu.cn (L.Z.); huan2268608512@163.com (H.S.); 3Key Laboratory of Crop Ecology and Molecular Physiology, Fujian Agriculture and Forestry University, Fuzhou 350002, China; 4State Key Laboratory of Proteomics, Beijing Proteome Research Center, Beijing 102206, China

**Keywords:** *Edwardsiella tarda*, acetylome, succinylome, antibiotic resistance

## Abstract

The antibiotic resistance of *Edwardsiella tarda* is becoming increasingly prevalent, and thus novel antimicrobial strategies are being sought. Lysine acylation has been demonstrated to play an important role in bacterial physiological functions, while its role in bacterial antibiotic resistance remains largely unclear. In this study, we investigated the lysine acetylation and succinylation profiles of *E. tarda* strain EIB202 using affinity antibody purification combined with LC-MS/MS. A total of 1511 lysine-acetylation sites were identified on 589 proteins, and 2346 lysine-succinylation sites were further identified on 692 proteins of this pathogen. Further bioinformatic analysis showed that both post-translational modifications (PTMs) were enriched in the tricarboxylic acid (TCA) cycle, pyruvate metabolism, biosynthesis, and carbon metabolism. In addition, 948 peptides of 437 proteins had overlapping associations with multiple metabolic pathways. Moreover, both acetylation and succinylation were found in many antimicrobial resistance (AMR) proteins, suggesting their potentially vital roles in antibiotic resistance. In general, our work provides insights into the acetylome and succinylome features responsible for the antibiotic resistance mechanism of *E. tarda*, and the results may facilitate future investigations into the pathogenesis of this bacterium.

## 1. Introduction

It is well-known that protein post-translational modifications (PTMs) play vital roles in diverse physiological and pathological functions in eukaryotic and prokaryotic cells. PTMs include phosphorylation, acetylation, glycosylation, and succinylation [1,2]. Among these PTMs, lysine acetylation (Kac) and succinylation (Ksu) modifications have been reported to be widely distributed in bacterial cells and have been implicated in biological processes such as chemotaxis [3], DNA replication, stress response [4], cell signaling transduction [5], nutrient metabolism [4], and virulence [6,7]. For example, the lysine deacylations YmcA at the K64 site and GtaB at the K89 and K191 sites dramatically decreased the biofilm formation in *Bacillus subtilis* [8]. In another example, the enzymatic activity of acetyl-CoA synthetase (Acs) was reported to be negatively regulated by the succinylation modification of the K193 and K336 sites on Acs in *Mycobacterium tuberculosis* [9]. Those lines of evidence confirm the importance of both lysine acylation modifications for maintaining bacterial survival when responding to a complex environment. Therefore, it is necessary to determine the Kac and Ksu modification profiles of bacterial species as much as possible before further investigating the biological functions of these PTM proteins.

Aquaculture currently plays an important role in food security and is a major source of national income in developed and developing countries [10]. However, the high stocking density employed in intensive aquaculture production has led to high sensitivity of fish to a variety of bacterial pathogens; approximately 65% of fish fry and fingerlings die due to bacterial infectious diseases. This has threatened the development of the global economy and world food security [11,12]. These fish diseases have been caused by aquatic pathogens, and the overuse of antibiotics in aquaculture has led to the prevalence of multidrug-resistant strains, thus creating a threat to global human public health [13]. Therefore, a better understanding of the mechanism of antibiotic resistance is necessary. Recently, several protein lysine acylation modification profiles were analyzed in aquatic pathogenic bacteria such as *Vibrio alginolyticus*, *Vibrio parahaemolyticus*, and *Aeromonas hydrophila*, and these were demonstrated to play important roles in various biological functions [6,14,15,16]. Moreover, common lysine acylations such as Kac and Ksu modifications were reported to be involved in bacterial antibiotic resistance, as well. For example, Fang et al. proposed that the deacetylation of the K413 site of *Escherichia coli* PykF led to the bacterium becoming sensitive to ampicillin, polymyxin B, and kanamycin by increasing the enzyme activity [17]. In addition to this, the accumulation of succinyl-CoA in the sucC mutant was reported to increase the methicillin-resistant *Staphylococcus aureus* (MRSA) succinylome and thereby increase the susceptibility to beta-lactam antibiotics [18]. Given the importance of bacterial antibiotic resistance in aquaculture, it is necessary to carry out large-scale identification and comparison of Kac and Ksu modification profiles in more aquatic pathogens.

*Edwardsiella tarda* is a well-known aquatic pathogenic bacterium typically isolated from animals that inhabit freshwater and marine environments [19]. It can infect a variety of hosts, including fish, amphibians, reptiles, birds, and mammals (including humans), and has become an enormous threat to many economically important fish species worldwide [20,21]. At present, to control infections, approximately 20 kinds of antibiotics are used. However, unfortunately, the overuse of antibiotics leads to the prevalent resistance to the pathogen. It was reported that the resistance of *E. tarda* to ampicillin ranges from 66 to 87.5%, and is 21–75% for oxytetracycline, thus generating interest in the resistance mechanism among those seeking to develop novel antimicrobial strategies [22]. Recent research reported that outer membrane proteins including the efflux pump, biofilm formation, and biosynthesis of fatty acids are associated with antibiotic-resistant *E. tarda* [22,23]. However, it is still unclear whether lysine acylation in *E. tarda* affects bacterial resistance to antibiotics.

In this study, we used high-affinity purification combined with LC-MS/MS technologies to compare the Kac and Ksu profiles of the *E. tarda* EIB202 strain. Further bioinformatic analysis showed that both PTMs are enriched in multiple metabolic biological processes such as the citrate cycle, pyruvate metabolism, biosynthesis of antibiotics, and carbon metabolism. Moreover, several Kac or Ksu proteins were identified to be involved in bacterial antimicrobial resistance (AMR). Here, we provide novel insights into the relationship between lysine acylation and the antibiotic resistance of *E. tarda*; the results may facilitate future investigations into the pathogenesis of this bacterium.

## 2. Results and Discussion

### 2.1. Proteomic Analysis of Lysine Acetylation and Succinylation in E. tarda

In nature, bacteria need to respond rapidly to adapt to changing environmental stresses such as the local temperature, osmotic pressure, redox potential, pH, nutrient availability, and host immune responses [24]. Lysine is a residue that can be modified by a variety of chemical groups via glutarylation, malonylation, propionylation, acetylation, or succinylation [25]. Among these lysine acylation modifications, lysine acetylation (Kac) and succinylation (Ksu) are both common PTMs that are well-distributed in prokaryotic cells [5]. To date, Kac and Ksu proteins have been reported to be involved in diverse biological processes such as intracellular metabolism, virulence, AMR, quorum sensing, and chemotaxis in many bacterial species [15,26,27,28], whereas their distributions and characteristics in *E. tarda* are unknown. In that context, using a combination of Kac and Ksu affinity enrichment and LC-MS/MS, we provide a comprehensive view of the lysine acetylome and succinylome in *E. tarda* EIB 202 in this paper. Through our work, a total of 1511 Kac sites from 589 proteins and 2353 Ksu sites from 692 proteins were identified. As shown in Figure 1A, the mass error of most acylation-modified peptides followed a normal distribution, ranging from −5 to 5 ppm with a mean near zero, indicating the expected error control from the MS dataset. Most of the enriched lysine-acylated peptide lengths were in the range of 7–23 amino acids, accounting for 94.77% and 93.52% of Kac and Ksu proteins (Figure 1B), respectively.

The identified Ksu (692 proteins) and Kac proteins (588 proteins) accounted for 19.3% and 16.4% of the total proteins (3580) in *E. tarda*, respectively. When compared to the bacterial acetylome and succinylome of other animal pathogens studied, the percentage of Kac proteins was lower than those of *V. alginolyticus* (27.1%) [14] and *A. hydrophila* (24.6%) [16] but was higher than in *V. parahemolyticus* (13.6%) and *V. mimicus* (15.5%) [29]. The percentage of Ksu proteins was lower than in *E. coli* (23.91%) [30] but was higher than in several species of bacteria such as *A. hydrophila* (16.0%) [28], *V. parahemolyticus* (13.3%) [15], *M. tuberculosis* (17.07%) [31], and *Pseudomonas aeruginosa* (10.5%) [32]. Of the identified Kac/Ksu proteins, approximately 40–50% contained a single PTM site, which was a little higher than the ~40% in *A. hydrophila* [28], and the number of modified proteins was decreased, with PTM sites significantly increased. As shown in Figure 1C, 51.1% of the 589 Kac proteins had one lysine-acetylated site; 20.7, 9, 4.07, 3.73, and 11.37% of the Kac proteins had two, three, four, five, or more than five acetylated sites, respectively. For the lysine-succinylation modification, 41.76% of the 692 Ksu proteins had one lysine-succinylated site; 19.08, 9.54, 8.53, 4.91, and 16.18% of the proteins had two, three, four, five, or more than five succinylated sites, respectively. Most notably, RpoC (D0Z9T3), the DNA-directed RNA polymerase beta subunit, possessed the highest numbers of succinylated (32) and acetylated (19) sites. Intriguingly, there were 13 sites on RpoC that were both acetylated and succinylated, indicating that these sites may be involved in crosstalk. RpoC and its homologous proteins are known to be associated with bacterial multidrug resistance. For example, the *rpoC* mutation affects daptomycin resistance in *S. aureus*, rifampin resistance in *M. tuberculosis*, and cefuroxime in *B. subtilis* [33,34,35]. In addition, several other proteins exhibited high abundances of both succinylated and acetylated sites, including the biosynthesis of antibiotic proteins D0ZH73 (18 Ksu and 10 Kac sites), D0ZCM8 (16 Ksu and 11 Kac sites), and D0ZCY4 (20 Ksu and 11 Kac sites). Detailed information concerning the identified peptides and matched proteins is presented in Appendix A.

Furthermore, homologs for at least nine Kac or Ksu *E. tarda* proteins have been reported to be related to antibiotic resistance in other bacterial species (Table 1). For example, there are at least three Kac sites at K76, K413, and K445 of pyruvate kinase PykF in *E. coli*, and the deacetylation of Lys413 in PykF was found to contribute to bacterial sensitivity to antibiotics such as ampicillin and polymyxin B [17]. These proteins were also found to be lysine acetylated or succinylated in our *E. tarda* data, though the modification sites might have differed, suggesting that lysine acylated proteins may have unique characteristics in *E. tarda* antibiotic resistance. Taken together, the results indicated that the Kac and Ksu modifications were well-distributed in *E. tarda* and that they may play important roles in bacterial biological functions, including antibiotic resistance.

### 2.2. Identification of Kac and Ksu Motifs in E. tarda

To identify both acylated sequence motifs, we used the Motif-X software, with amino acid sequences that comprised at least 21 amino acids, from −10 to +10 residues surrounding the Kac and Ksu sites. Five conserved motifs were significantly enriched in Kac or Ksu peptides, including V.K…..K, D…..RK, KH, KL, and EKL. Among these, V.K…..K and D…..RK were unique for succinylated peptides, and the KH, KL, and EKL motifs were enriched in both acylated peptides (Figure 2). The highest enrichment of leucine (L) was observed at the +1 position. The frequency of the KsuL motif was much higher than those of other motifs, and the KsuH motif was one of the top two enriched motifs. Moreover, KsuH, KsuL, and EKsuL were significantly overrepresented among acetylation peptides. The motif analysis revealed that both motifs (Ksu/acH and Ksu/acL) may be functionally important for acetylation, and all five conserved sites may be functionally important for succinylation in *E. tarda*. Motif KacH has been reported to be conserved and is widespread in other bacterial species such as *Streptococcus mutans* [38], *S. pneumoniae* [39], *Mycobacterium abscessus* [40], *Saccharopolyspora erythraea* [41], and *M. tuberculosis* [42]. Motif KsuL has been reported to be enriched in *B. subtilis* [43,44], and Motif KacL has been reported in *V. mimicus* [29]. However, the other motifs, Ksu H, EKsu L, EKac L, V.Ksu ….K, and D…..R.Ksu, have rarely been identified in other bacteria.

### 2.3. Functional Annotation of Kac and Ksu Proteins in E. tarda

To better understand the potential function of the identified proteins that were associated with both acylation-modified sites, subcellular localization prediction and GO and KEGG analyses were performed in this study. We first used BUSCA software to predict the subcellular localization of identified acylation-modified proteins. Both types of proteins primarily belonged to the cytoplasm in *E. tarda* (521 Kac and 564 Ksu proteins, accounting for 88.46 and 81.5% of the total PTM proteins, respectively). These results were consistent with previous studies indicating that both PTMs appear to be well-represented in intracellular metabolic pathways and protein biosynthesis [31]. Our finding of more than 30 ribosomal PTMs, including 30S and 50S ribosomal subunits in the cytoplasm, further confirmed that PTM contributed to protein translation. Other PTM proteins may be related to transport and antibiotic resistance in cell envelope proteins, including 7.47, 3.57, and 0.51% of the Kac proteins that were distributed in the plasma membrane, extracellular space, and outer membrane, and 9.54, 8.67, and 0.29% of the Ksu proteins that were dispersed in the plasma membrane, extracellular space, and outer membrane, respectively (Figure 3A). Interestingly, several chaperones, including GroEL, DnaK, DnaJ, and SurA, were located in the cytoplasm and extracellular space. Previous transcriptional and proteomics results have reported that several chaperones are involved in bacterial drug resistance. For example, GroEL and DnaK affect the regulation of antibiotic resistance by inhibiting cytosolic protein misfolding to increase the bacterial tolerance to aminoglycoside antibiotics and susceptibility to fluoroquinolones [45,46]. Although several lines of research have identified the Kac or Ksu modifications on Dnak or GroEL, the actual roles of both acylation modifications in affecting chaperones are still elusive [47].

Gene ontology analysis was also conducted to investigate the functional enrichment of Kac and Ksu proteins. In the biological process (BP) category, Kac proteins were significantly enriched in cellular and primary and organonitrogen compound metabolic processes, whereas the Ksu proteins tended to be involved in gene expression and primary and organonitrogen compound metabolic processes (Figure 3B). Similarly, in the cellular component (CC) category, most of the identified lysine-acetylated and -succinylated proteins were prevalent in the cell, cell part, cytoplasm, intracellular part, and intracellular categories (Figure 3C). Consistent with previous research, the identified Kac proteins were enriched in metabolic processes and cell locations similar to BW25113 and *V. mimicus* [29,48]. When compared to Kac proteins, the significantly enriched categories were quite different for Ksu proteins in the molecular function (MF) category. Kac proteins were associated with several substrates, such as in nucleotide, organic cyclic, small-molecule, nucleoside phosphate, and heterocyclic compound binding, while organic cyclic compound and ion binding were enriched in Ksu proteins (Figure 3D).

### 2.4. KEGG Analysis of Cross-Talking Proteins in Kac and Ksu in E. tarda

In our previous study, we demonstrated that some proteins potentially engaged in crosstalk in lysine acetylation and succinylation modification and that they played important regulatory roles in *A. hydrophila* [16]. Hence, it was necessary to assess the characteristics of cross-talking proteins in *E. tarda*. There were 437 proteins comprising 948 peptides that overlapped at the same lysine residues in *E. tarda*, suggesting that lysine succinylation and acetylation frequently occurred. Furthermore, KEGG enrichment analysis showed that many overlapping cross-talking proteins were significantly associated with carbon metabolism, ribosomes, RNA degradation, pyruvate metabolism, glycolysis/gluconeogenesis, and the TCA cycle. Several metabolic pathways such as pyrimidine metabolism, oxidative phosphorylation, and lysine degradation were found to be unique to lysine acetylation, whereas pyruvate metabolism, RNA degradation, and amino sugar and nucleotide sugar metabolism were unique to succinylation (Figure 4). These results indicate that the two PTMs may have their own characteristics and that this factor needs to be further explored.

### 2.5. Prediction of Protein-Protein Interaction Networks of Kac and Ksu Proteins in E. tarda

We constructed protein-protein interaction (PPI) networks involving the observed Kac and Ksu proteins based on the STRING database using Cytoscape enriched by KEGG pathways. At least five highly metabolic pathways of PTM proteins were enriched in the global PPI network, including ribosomes, aminoacyl-tRNA biosynthesis, the pentose phosphate pathway, RNA degradation, and oxidative phosphorylation. The high-ranking interaction clusters are shown in Figure 5; there were 46 ribosomal subunit proteins from overlapped proteins between lysine acetylation and succinylation that were highly enriched. Previous research has documented that ribosome subunits may be involved in antibiotic resistance. For example, 10 ribosome subunits in *A. hydrophila* biofilms were significantly increased in response to chlortetracycline. The increased levels of these ribosomal proteins may help to relieve the pressure from tetracycline attacks on translation processes [49]. As another example, Klitgaard et al. reported that mutations of the bacterial ribosomal protein L3 could reduce the susceptibility to tiamulin or linezolid, indicating the important role of ribosome subunits in bacterial antibiotic resistance [50]. Moreover, 23 acylated modified proteins, including Kac protein CysS and Ksu proteins GlyQ, ProS, and LeuS, were found to be associated with aminoacyl-tRNA biosynthesis, which may affect bacterial antibiotic resistance [51]. As expected, the PPI analysis indicated that lysine acetylation and succinylation in *E. tarda* take part in multiple metabolic pathways, including the pentose phosphate pathway, the TCA cycle, and pyruvate metabolism. Several lines of research have demonstrated that central metabolic pathways of bacteria such as the TCA cycle and glycogenesis/glycolysis, as well as multiple related metabolites, are involved in the regulation of bacterial antibiotic susceptibility [27,52]. Hence, metabolic reprogramming by controlling the status of Kac or Ksu modification may be a novel strategy by which *E. tarda* regulates bacterial resistance.

### 2.6. Kac/Ksu AMR Proteins in E. tarda

We then further asked whether the common lysine acylations modify AMR proteins. By homologous searching of an AMR gene database (CARD), we found that a total of 66 PTM proteins, including 45 Kac and 56 Ksu proteins with 35 common proteins in *E. tarda*, belonged to AMR proteins, and that the majority could be used to construct a complicated protein-protein interaction (PPI) network (Figure 6, Appendix A). In the network, we found that the majority of acylated AMR proteins interacted directly, and approximately 95% of the proteins were involved in antibiotic resistance through bacterial efflux pump transportation. In addition, other proteins were involved in multiple physiological processes to regulate bacterial antibiotic resistance, such as the Sec-dependent secretion system, the ABC transport pathway, reduced permeability to antibiotics, antibiotic inactivation, and altered antibiotic targets. For efflux pump transportation, several antibiotic efflux transporter-related proteins such as AcrA/AcrB, RstA/RstB, OppD, D0Z9L4, and D0Z9I3 were reported to be involved in the efflux pump controlling antibiotic injection into cells in other bacterial species [53]. For example, the AcrAB-TolC system that belongs to the RND (resistance-nodulation-cell division)-type transporter is a typical multidrug resistance (MDR) efflux pump in Gram-negative bacteria [54,55]. SecD, which is involved in protein translocation, has been shown to play a significant role in β-lactam resistance in *S. aureus* [56]. Additionally, some MDR-related proteins were also found to be Kac- or Ksu-modified in this study. Among these, OppD is a major component of the β-lactam antibiotic resistance of *S. agalactiae* [57]; ArnA is a bifunctional polymyxin resistance protein targeted to peptide antibiotics by altering the antibiotic target of the host [58]. Moreover, the protein EmrA is involved in the regulation of multidrug resistance in various bacterial species [59,60]. Thus, we speculate that acetylation and succinylation may play critical roles in regulating bacterial antibiotic resistance in *E. tarda*.

## 3. Materials and Methods

### 3.1. Bacterial Strains and Protein Extraction

The *E. tarda* EIB202 strain was kindly provided by Prof. Xuanxian Peng of Sun Yat-Sen University (Guangzhou, China). A single clone of the *E. tarda* strain was cultured overnight in Luria Bertani (LB) medium and then diluted at a ratio of 1:100 in 100 mL of LB at 30 °C. The cells were harvested until the OD at 600 nm reached 1.0 and then were washed three times with pre-cooled PBS (pH 7.5). The cell pellets were resuspended in lysis buffer (8 M urea, 2 mM EDTA, 50 mM Tris-HCl pH 8.5, protease inhibitor) and then ultrasonically disrupted by sonication on ice. After sonication, the proteins in the supernatant were collected through centrifugation at 12,000× *g* for 10 min at 4 °C.

### 3.2. Protein Digestion and Immunoaffinity Enrichment of Lysine-Acetylated and -Succinylated Peptides

In this study, protein samples were digested to peptides. The detailed protocol was as follows: 20 mg protein was reduced with 10 mM dithiothreitol for 2 h at 37 °C. Then, the protein was alkylated under 50 mM iodoacetamide for 30 min in the dark at room temperature. Five times the volume of ddH_2_O was added to dilute the urea concentration to 1 M, and then digested with trypsin at a ratio of 1:20 for 16 h at 37 °C. After digestion, the peptides were desalted using an SPE C18 column and then vacuum-lyophilized. The lysine-acetylated and succinylated peptides were enriched by immunoaffinity analysis using agarose-conjugated anti-acetyllysine and anti-succinyllysine antibody (PTM Biolabs Inc., Hangzhou, China), respectively, as previously described [28,30]. Briefly, the digested peptides were incubated with anti-acetyllysine or anti-succinyllysine agarose beads overnight at 4 °C in NETN buffer (100 mM NaCl, 1 mM EDTA, 50 mM Tris-HCl, and 0.5% NP 40, pH 8.0). After incubation, the beads were washed with NETN buffer four times and with ddH_2_O two times. Then, the modified peptides were eluted with 0.1% formic acid (FA) and desalted with C18 ZipTips (Millipore, Burlington, MA, USA) before being analyzed with reversed-phase liquid mass RPLC-MS/MS.

### 3.3. Protein Analysis by LC-MS/MS

Proteins were identified using the Thermo Q Exactive HF mass spectrometer (Thermo Fisher Scientific, Waltham, MA, USA) as previously described [48]. Digested peptides were dissolved in water containing 0.1% FA and separated by the RPLC C18 capillary reversed-phase analytical column (Thermo Fisher Scientific, Waltham, MA, USA) with an 80 min 7–20% acetonitrile (ACN)/water gradient containing 0.1% FA and a 24 min 20–32% ACN/water gradient containing 0.1% FA at a flow rate of 600 nL/min on the EASY-nLC 1000 system. The eluted peptides were further ionized and sprayed into the nanospray-ionization source followed by tandem mass spectrometry (MS/MS) in Q Exactive HF. The raw data files obtained from the MS analysis were processed using Maxquant (v.1.6.3.0) with the false discovery rate (FDR) <1%; the identified Kac/Ksu sites had a localization probability >0.75 and a score >35.

### 3.4. Bioinformatic Analysis

The Gene Ontology (GO) annotations and Kyoto Encyclopedia of Genes and Genomes (KEGG) pathways were enriched by KOBAS (http://kobas.cbi.pku.edu.cn, accessed on 2 October 2021) [61]. The protein subcellular localization was predicted with BUSCA (https://busca.biocomp.unibo.it/, accessed on 2 October 2021), and the motif analysis was performed by Motif-X using amino acid sequences that were composed of at least 20 amino acids within ±10 residues of the lysine acylated sites (http://motif-x.med.arvard.edu, accessed on 3 October 2021) [62,63]. Protein-protein interaction networks were constructed by STRING version 11.5 online software and visualized with Cytoscape v3.7.1 (http://string-db.org, accessed on 5 October 2021) [64,65]. The Comprehensive Antibiotic Research Database (CARD, http://arpcard.mcmaster.ca, accessed on 15 October 2021) was used to predict antibiotic-resistant modified proteins [66].

## 4. Conclusions

In this study, we investigated the lysine acetylation and succinylation profiles of *E. tarda* strain EIB202 using affinity antibody purification combined with LC-MS/MS. A total of 1511 lysine-acetylation sites were identified on 589 proteins, and 2346 lysine-succinylation sites were further identified on 692 proteins of this pathogen. Additionally, five conserved motifs were found, Kac/_suc_H_(+1)_, Kac/_suc_L_(+1)_, E_(−1)_Kac/_suc_L_(+1)_, V_(−2)_Ksu K_(+10)_, and D_(−8)_R_(−1)_Ksu, in Kac and Ksu modifications. Further bioinformatic analysis showed that both PTMs were enriched in cellular, primary, and organonitrogen compound metabolic biological processes. Moreover, at least 45 Kac and 56 Ksu proteins were involved in bacterial AMR. In comparisons of the obtained lysine acetylome and succinylome, 948 peptides of 437 proteins were found to overlap and to be associated with multiple metabolic pathways. In summary, our results provide in-depth *E. tarda* lysine acetylome and succinylome profiles, and for the first time, reveal the role of crosstalk between lysine acetylation and succinylation and its potential impact on bacterial antibiotic resistance.

## Figures and Tables

**Figure 1 antibiotics-11-00841-f001:**
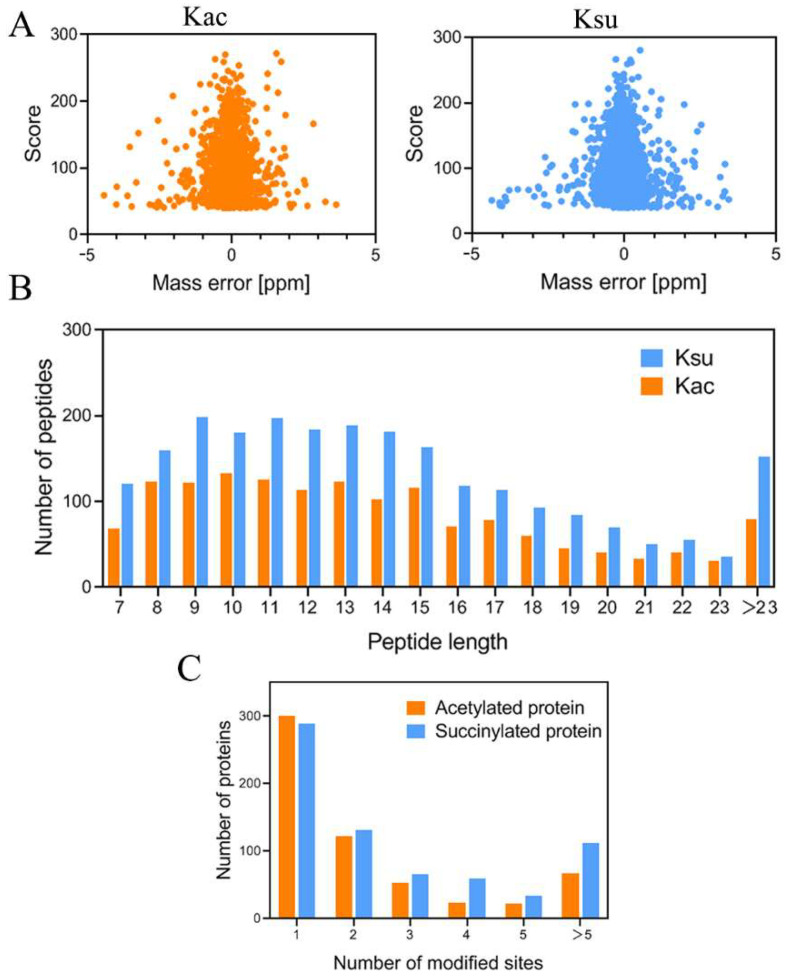
Profile of *E. tarda* lysine acetylation and succinylation proteome. (**A**) Distributions of mass errors for Kac and Ksu peptides. (**B**) Distributions of Kac and Ksu peptides based on their lengths. (**C**) Distributions of modified sites in Kac and Ksu proteins.

**Figure 2 antibiotics-11-00841-f002:**
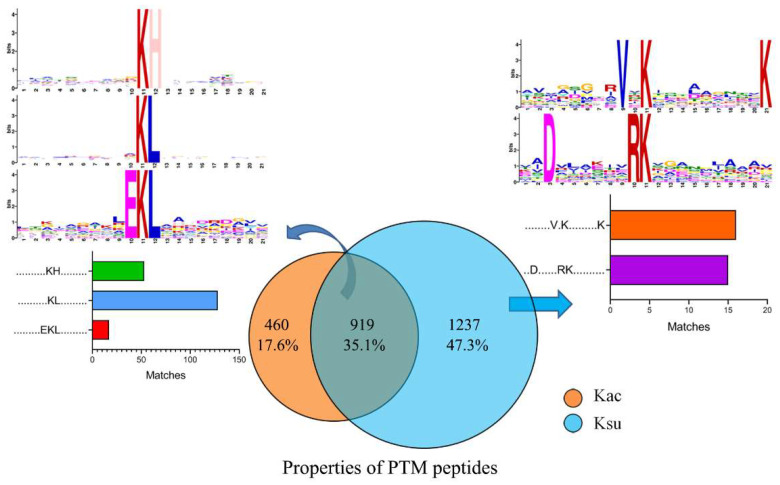
Motif analysis of acetylated and succinylated peptides in *E. tarda*. Venn diagram shows the overlapping peptides between the PTMs.

**Figure 3 antibiotics-11-00841-f003:**
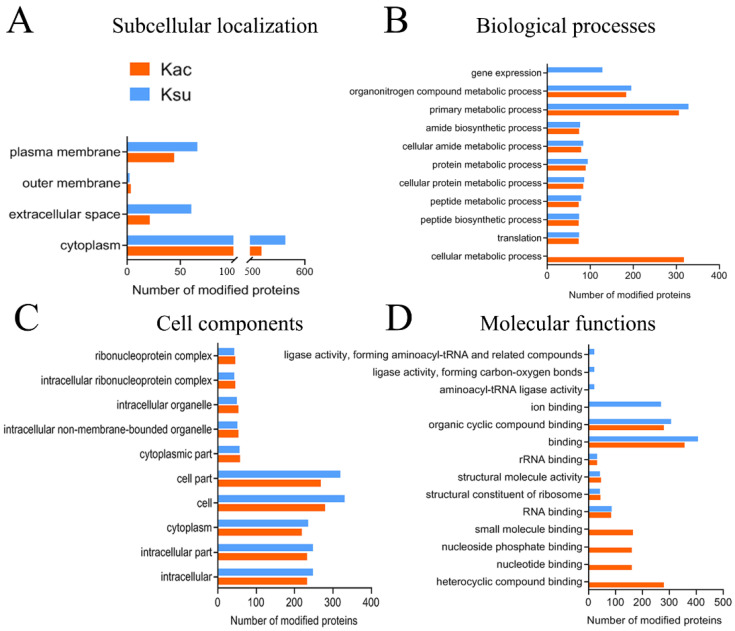
Subcellular localization prediction and GO annotation analysis of acetylated/succinylated-lysine proteins in *E. tarda* strain EIB 202. (**A**) Subcellular localization of the identified Kac and Ksu proteins. (**B**–**D**) GO annotation analysis of the identified Kac and Ksu proteins in terms of biological process, cell components, and molecular function.

**Figure 4 antibiotics-11-00841-f004:**
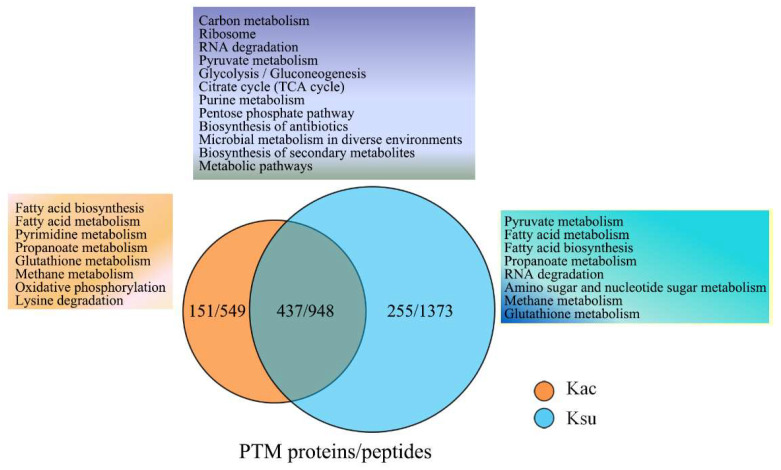
KEGG pathway overlap between lysine acetylation and succinylation in *E. tarda*. Venn diagram displaying the overlapping peptides and proteins between the two PTMs. The orange, purple, and blue regions represent the KEGG pathways related to unique acetylated, overlapping, and unique succinylated proteins, respectively.

**Figure 5 antibiotics-11-00841-f005:**
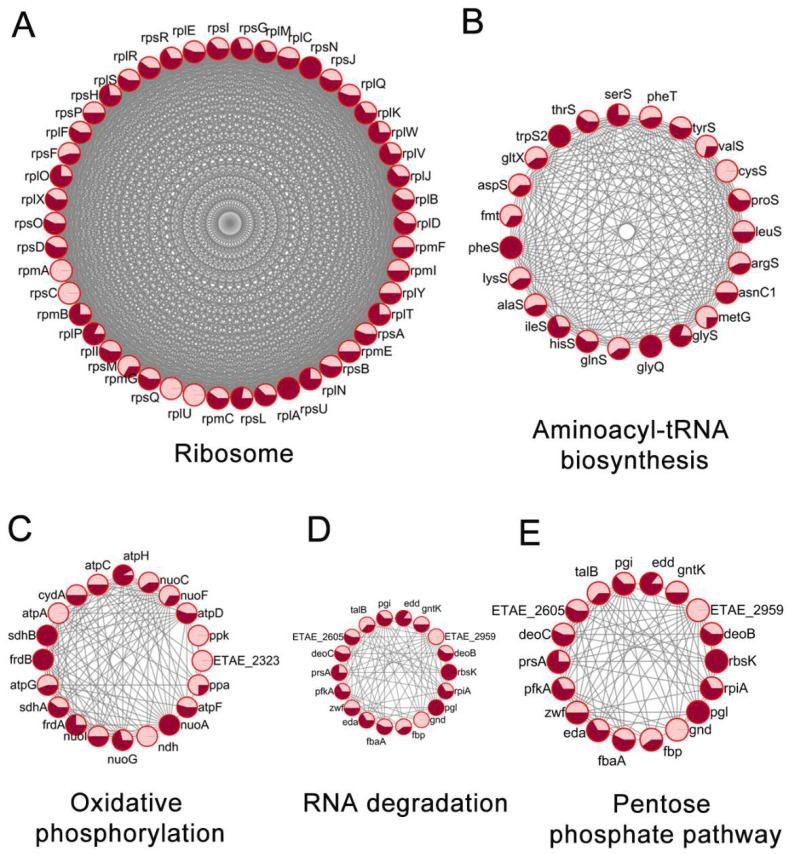
Prediction of protein-protein interaction networks of lysine acetylation and succinylation proteins in *E. tarda*. There were five highly metabolic pathways of PTM proteins were enriched in the global PPI network, such as, (**A**) ribosome, (**B**) aminoacyl-tRNA biosynthesis, (**C**) oxidative phosphorylation, (**D**) RNA degradation, (**E**) pentose phosphate pathway. Different colors display different modifications; the pink and red cycles represent the proportions of Kac and Ksu modifications in single proteins, respectively.

**Figure 6 antibiotics-11-00841-f006:**
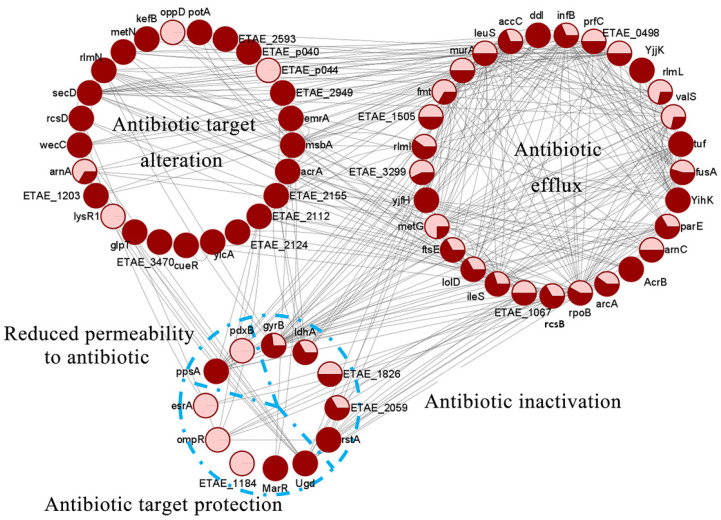
Analysis of protein-protein interaction networks of 66 AMR proteins in *E. tarda*. The pink and red cycles represent the proportions of Kac and Ksu modifications in single AMR proteins, respectively.

**Table 1 antibiotics-11-00841-t001:** The PTMs of AMR proteins in previous reports and in this study.

Protein	Antibiotic	Lysine Modified Type	Lysine Acylated Sites	Reference
PykF	Ampicillin; polymyxin B; kanamycin	Acetylation	**413**	[17]
		Acetylation	68; 56; 382; 319; 173; 434; 13; 286	This study
		Succinylation	68; 56; 208; 382; 319; 173; 175; 434; 13; 3; 5; 272; 286; 266; 413	This study
KatG	Isoniazid	Succinylation	557; 143; 600; 356; 310; 590; 688; 554; 433	[31]
		Succinylation	11	This study
GyrA	Fluoroquinolone	Succinylation	325; 49; 319; 224; 245	[36]
		Acetylation	465; 76	This study
		Succinylation	270; 657; 465; 76; 754	This study
MetRS	Chloramphenicol	Succinylation	362; 388	[37]
		Acetylation	148; 602; 407	This study
		Succinylation	602	This study

The bold value in the table represents the same lysine modification sites in this study and a previous report.

## Data Availability

The mass spectrometry proteomics raw datasets generated for this study have been deposited to the ProteomeXchange Consortium via the iProX partner repository with the dataset identifier PXD033818 (http://proteomecentral.proteomexchange.org/cgi/GetDataset?ID=PXD033818, accessed on 9 May 2022).

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
