# Peer review of "Acetylome and Succinylome Profiling of *Edwardsiella tarda* Reveals Key Roles of Both Lysine Acylations in Bacterial Antibiotic Resistance"

_antibiotics, 2022, doi:10.3390/antibiotics11070841_

Round 1
Reviewer 1 Report
The manuscript of Fu and co-authors deals with the relation between the bacterial antibiotic resistance and the post-translational modification. I found the paper well written with a detailed description the experimental methods and conclusions well supported by the results. I suggest that the authors consider the following minor revision in preparing the final manuscript:
1. line 25, please provide the entire name for PTM and TCA
2. double references are reported sometimes togheter somtimes not (ex. line 81 and line 77)
3. line 118, P aeruginos has to be written extended the first time
4. line 1321 and 132, the name of bacteria should be abbreviated
5. Figure 3, pannel A, Subcellular localization, a l is misssing
As regard the introduction I suggest the authors to introduce, if possibile, some additional information regardin the ressitance profile pf Edwardsiella tarda.
Author Response
Dear antibiotics Editor,
Thank you for your letter and for the reviewers’ comments concerning our manuscript entitled “Acetylome and succinylome profiling of Edwardsiella tarda reveals key roles of both lysine acylations in bacterial antibiotic resistance” (antibiotics-1766404). We are also grateful for the comments from the reviewer on our manuscript. Those comments are all valuable and very helpful for revising and improving our paper. We carefully considered every comment, and made cautious revisions accordingly. Based on their suggestions, we have answered all of the questions in detail point by point, to the referees’ comments. The reviewer gave us those valuable suggestions may help us to have a modification ever before.
We hope the revised paper would satisfy you. And we are looking forward to hearing from you.
Please note that all the modifications to the revised manuscript are in red-colored. The answers to the Reviewer’ comments are also in red-colored in the response letter.

Reviewer 2 Report
Acetylome and succinylome profiling of Edwardsiella tarda re-2 veals key roles of both lysine acylations in bacterial antibiotic resistance
Comments:- the manuscript is well coated, informative and well written. However few corrections need to be addressed
Major:-
Please check reference number 20, 21, 22, 23 in text in Page2
Please check font of “Results and Discussion” in heading in page 2
In text the peptide is represented as “Kac” and “Ksu” while in figure, it is mentioned as “Kace” and “Ksuc” respectively. Please check
Some references in text are italics. Please unitalic the references to make it consistent in manuscript
Figure 5, some labellings overlap each other, making the readability unclear. Please check.
Minor:-
Pg1
Line 25:- In Abstract section, please provide the full form of PTMs.
Line 38:- please change “lysine acetylation (Kac) and succinylation (Ksu) modifications” to “lysine acetylation (Kac), and succinylation (Ksu) modifications”
Line 43:- please unitalicize reference number 8 in text.
Pg3
Line 99, please change “To date, Kac and Ksu proteins have” to “Till date, Kac and Ksu proteins have”
Line 127, please change “RNA polymerase subunit beta” to “RNA polymerase beta subunit” if possible
Line 134, please make the representation of protein and sites consistent in “proteins D0ZH73 134 (18 Ksu and 10 Kac sites), D0ZCM8 (16, 11), and D0ZCY4 (20, 11).”
Pg5
Line 167, please unitalic the references
Pg7
Line 258, Please change “glycogenesis\glycolysis as well as multiple” to “glycogenesis/glycolysis as well as multiple”
Pg9
Line 311, please elaborate/rephrase “then trypsin (20:1) was added for digestion for 16 h at 37°C” for better understanding
Pg12-16
References:- please try to replace references before 2010 with some recent studies, if possible
Author Response

(The authors gave the same response as above.)

Reviewer 3 Report
Dear authors,
I have read very carefully your work, which I find very interesting and very topical.
Identifying the role of crosstalk between lysine acetylation and succinylation and its potential impact on bacterial antibiotic resistance will also pave the way for the discovery of new antibiotic compounds in the fight against multidrug-resistant bacteria.
The article is well written in a comprehensive manner. My recommendation is to publish as it is.
Author Response
Dear antibiotics Editor,
Thank you for your letter and for the reviewers’ comments concerning our manuscript entitled “Acetylome and succinylome profiling of Edwardsiella tarda reveals key roles of both lysine acylations in bacterial antibiotic resistance” (antibiotics-1766404). We are also grateful for the comments from the reviewer on our manuscript.